# A Sustainable Diet for Tambaqui Farming in the Amazon: Growth Performance, Hematological Parameters, Whole-Body Composition and Fillet Color

**DOI:** 10.3390/ani14081165

**Published:** 2024-04-12

**Authors:** Francisco de Matos Dantas, Yasmin Moreira de Souza, Thiago Macedo Santana, Driely Kathriny Monteiro dos Santos, Flávio Augusto Leão da Fonseca, Ligia Uribe Gonçalves

**Affiliations:** 1Post-Graduate Program in Animal Science and Fishing Resources—PPGCARP, Federal University of Amazonas, Manaus 69067-055, AM, Brazil; dantasfm3@gmail.com (F.d.M.D.); zootecthiago@gmail.com (T.M.S.); 2Agricultural Sciences College, Federal University of Amazonas, Manaus 69067-055, AM, Brazil; yasminmoreiraa570@gmail.com; 3The National Institute for Amazonian Research, Manaus 69060-001, AM, Brazil; driely.monteiro@gmail.com; 4Federal Institute of Education, Science and Technology of Amazonas, Manaus 69083-000, AM, Brazil; flavio.fonseca@ifam.edu.br

**Keywords:** aquafeed, alternative ingredients, *Colossoma macropomum*, *Hermetia illucens*, local ingredients

## Abstract

**Simple Summary:**

This study focused on developing feed based on locally sourced ingredients for tambaqui farming in the Amazon. We tested diet formulations with varying levels (0% to 100%) of defatted black soldier fly larvae meal as a replacement for traditional fish meal, combined with cassava by-products (tuber residues, peel, and leaves). A traditional diet was used as the control. Neither diet rejection nor mortality were observed. Fish fed all the experimental diets presented similar feed conversion and protein efficiency rates. However, these results were worse than those observed in the fish fed a traditional diet. There were no significant differences in the overall body composition of the fish, but those fed cassava by-products developed yellow-colored fillets due to the carotenoids in the cassava leaves. The black soldier fly larvae meal and cassava by-products are a viable way to support sustainable aquaculture in the Amazon; however, we recommend further research to optimize the diet formulation, particularly a reduction in the proportion of cassava leaves, in order to improve digestibility and minimize the impact on fillet color.

**Abstract:**

The aim of this study was to produce feed based on locally sourced ingredients for tambaqui farming in Amazon. Diets were formulated with increasing levels (0, 25, 50, 75 and 100%) of defatted black soldier fly larvae meal (BSFL) as a replacement for fish meal (FM), and cassava by-products in the same proportion (tuber residues, peel and leaves). A conventional diet (CO) was used as the control. Juvenile tambaqui (24.61 ± 1.14 g) were housed in 24 tanks in a recirculation aquaculture system. Neither diet rejection nor mortality were observed. Fish fed cassava by-products showed similar feed conversion rates (FCR 1.76); however, these values were worse than those observed in fish fed the CO (FCR 1.33). No differences were observed in the whole-body composition of the fish. The fillets of fish fed cassava by-products had a yellow color due the carotenoids present in the leaves. Dietary BSFL and cassava by-products can contribute to the sustainability of Amazonian aquaculture. Further studies with a lower proportion of cassava leaves in the diet formulation are recommended so as to ensure enhanced diet digestibility and less impact on the color of the fillets.

## 1. Introduction

Aquaculture in the Brazilian Amazon is primarily (97.17%) composed of small properties of up to 5 hectares that are mainly engaged in cultivation of tambaqui (*Colossoma macropomum*) [1]. The pre-eminence of tambaqui in Brazilian fish farming is attributed to its compatibility with artificial propagation techniques to obtain juveniles, a pronounced capacity for exploiting plankton as a nutritional resource, and their favorable productive response to commercially formulated feeds [2,3,4].

Despite being incorporated into the Brazilian federal government’s developmental strategy for their share of the Amazon, aquaculture in the region faces significant economic hurdles due, especially, to high production costs [1]. This is mainly due to the high cost of logistics to transport the raw materials and aquafeed from other regions. However, there are well-established production chains in the Amazon that generate significant volumes of by-products, which are frequently underutilized or improperly disposed of, resulting not only in considerable waste and financial loss but also adverse environmental impacts. These by-products have the potential for repurposing as components in aquafeed formulations and present an opportunity to enhance the sustainability and cost-efficiency of aquaculture practices in the area.

The by-products of the cassava (*Manihot esculenta*) production chain, including tuber residues, peel and leaves, constitute a rich source of starch, energy, and fiber, making them viable for the formulation of extruded fish feed. Meal from cassava tuber residues, a by-product of the grinding and sieving that occurs in the manufacture of cassava flour, contains approximately 82% starch and 15.61 MJ of gross energy/kg. This composition underscores its potential as an energy-dense ingredient suitable for inclusion in extruded feeds. Cassava starch is rich in amylopectin, which has a significant capability to expand pellets during the extrusion process, and results in the buoyancy of pellets [5].

The cassava peel is obtained during the cleaning of the cassava tuber and represents 20% of its weight. Its starch content contributes to the agglutination of all the dietary components during the feed extrusion process [6]. The extrusion and drying processes reduce the toxic compounds found in cassava by-products and disrupt the plant cell wall, which exposes the starch to the action of amylases [7], thus improving the nutritional availability of the cassava peel for monogastric animals [8].

Cassava leaves become a valuable nutritional resource once dried and devoid of the petiole, and have a crude protein content that can reach up to 28% [9], and the aerial part (upper third with leaves and petioles) of a 16-month-old plant contains protein levels that range from 7 to 13% [10]. The use of fresh cassava leaves in animal nutrition is limited due to the toxic elements, such as cyanogenic glycosides (linamarin and lotaustralin), present in their composition. However, processes, such as dehydration, grinding, and cooking (i.e., extrusion), are effective for significantly diminishing the toxic compounds found in the cassava [11,12].

Whole-fish meal from freshwater fish is one of the few protein ingredients available for animal nutrition in the Amazon; however, its constantly growing demand could become unsustainable for fishing and aquaculture in the medium and long term. An alternative to fishmeal is insect larvae, such as the black soldier fly (*Hermetia illucens*), which develops very well in high temperatures and high humidity, natural conditions throughout the year in the Amazon. Black soldier fly larvae (BSFL) recycle solid organic waste, resulting in larvae biomass with values of up to 42% crude protein and 30% lipids [13] and, when defatted, the protein content can reach 60% [14].

Our previous research has shown that tambaqui can be fed with full-fat BSFL to a replacement level of 50% of the commercial feed [15], and the full-fat or defatted BSFL can be added up to 15.75% [16] or up to 30%, respectively, in extruded feed without compromising tambaqui growth performance [17]. These studies were carried out with BSFL as a commercial feed substitute or as an ingredient in regular fish feed formulations, using ingredients such as soybean, corn, and wheat middlings. However, it is recognized that this diet could be applied in other regions around the world where cassava is grown and where BSFL can be cultivated. In the present study, our aim was to formulate an aquafeed using by-products from the local cassava production chain and BSFL meal replacing the traditional freshwater fish meal. 

## 2. Materials and Methods

The experiment was carried out at the Aquaculture Experimental Station of the Technology and Innovation Coordination (COTEI/INPA) (3°05′26.7″ S and 59°59′41.1″ W), Amazonas state, Brazil. This study was approved by the Ethics Committee for Research on the Use of Animals of the National Institute for Amazonian Research (INPA) under protocol No. 137/2022.

### 2.1. Ingredients and Experimental Diets

Cassava tuber residues (CT), cassava peel (CP) and cassava leaves (CL) were used as a source of starch, energy and fiber (Figure 1). The by-products from the cassava were collected on farmers’ properties in the municipality of Careiro, Amazonas, Brazil (Lago do Janauacá, Boa Vista Community, 3°29′19.89″ S and 60°16′08.64″ W) and transported to INPA. The by-products were dried in an oven with air circulation at 55 °C until reaching a moisture content of less than 10%. After drying, each raw material was ground in a hammer mill to a particle size of 1 mm. Whole-freshwater-fish meal (FM) and defatted BSFL meal were produced by the company Cyns (Piracicaba, SP, Brazil), and both were used as a protein source.

The raw materials were analyzed for their proximate composition [18] and gross energy, which was measured in a bomb calorimeter (IKA 2600, Werke GmbH & Co. KG, Staufen, Germany) (Table 1). The crude protein content of the BSFL meal was calculated using 5.6 as the correction factor, as recommended by [19], so as not to overestimate the protein value due to non-protein nitrogen from chitin.

The cassava leaves, BSFL and whole-freshwater-fish meal were analyzed for amino acid content. Samples of 100 mg of each ingredient were hydrolyzed with 6 M HCl at 110 °C for 24 h, followed by neutralization with 4 mL of 25% NaOH (*w*/*v*) and then cooled to room temperature. The mixture was then brought up to 50 mL volume with sodium citrate buffer (pH 2.2) and analyzed using an amino acid analyzer (1260 Infinity LCs (Agilent Technologies, Santa Clara, CA, USA). Tryptophan was determined using the colorimetric method [20], a standard curve of pure tryptophan (Merck KGaA, Darmstadt, HE, Germany) and detection at 590 nm with a spectrophotometer (DU-640 UV/Vis—Beckman Coulter, Basking Ridge, NJ, USA). Cystine and methionine were analyzed as cysteic acid and methionine sulfone via oxidation with performic acid for 16 h at 0 °C and neutralization with hydrobromic acid before hydrolysis (Table 2).

Five isoprotein and isoenergetic diets were formulated with equal proportions of cassava by-products, and with increasing levels of replacement of fish meal by BSFL meal (0; 25; 50; 75 and 100%) (Table 3). In addition to the five diets, a control diet was formulated with ingredients commonly used in commercial feeds used for tambaqui (Table 3).

The ingredients were mixed, ground, and extruded in a single screw extruder (INBRAMAQ, Model MX-80, São Paulo, SP, Brazil) with a 3 mm matrix, a motor amperage between 30 and 38 A, a screw frequency of 60 Hz and a knife frequency of 28.0 to 31.5 Hz. After extrusion, the pellets were dried in an oven with air circulation at 55 °C until reaching a moisture level below 10%.

### 2.2. Feeding Trial

Juvenile tambaqui (24.61 ± 1.14 g; 10.95 ± 0.26 cm; 85 days after hatching) raised in an earthen pond, fed with commercial feed (40% crude protein), were purchased from the Santo Antônio farm, Rio Preto da Eva, Amazonas, Brazil. Fish were randomly distributed in 24 polyethylene tanks with a useful volume of 150 L (n = 4; 20 fish/tank) in a completely randomized design. The fish were kept in a recirculation system with phytoremediation, constant aeration and a natural photoperiod (12/12). In the phytoremediation, effluent recirculates through a *Pistia stratiotes* tank, absorbing excess nitrogen and phosphorus, before returning to the experimental tanks (water flow 0.03 m^3^/s). The fish were fed four times a day (8:00, 11:00, 14:00 and 17:00) until apparent satiety for 60 days.

Water quality parameters (temperature 29.04 ± 0.89 °C; pH: 6.24 ± 0.29; dissolved oxygen 6.30 ± 0.66 mg L^−1^) were monitored daily using multiparameter probe (HI98196; HANNA^®^). Ammonia (0.33 ± 0.17 mg L^−1^) and nitrite (0.02 ± 0.01 mg L^−1^) levels were monitored weekly using colorimetric and titrimetric kits (Alfakit AT 101; Alfakit, Florianópolis, SC, Brazil). All parameters were within the comfort range for tambaqui [22].

The length and weight of the fish were recorded at the beginning and end of the experiment. At the end of the experiment, the fish were fasted for 24 h and anesthetized by immersing them in a benzocaine solution (100 mg L^−1^) in order to perform the biometry [23]. Three fish from each experimental unit were euthanized via spinal cord rupture, then frozen (−20 °C) for proximate composition analysis.

At the end of the experiment, the growth performance data were obtained by the following calculations: weight gain (WG, g) = final weight − initial weight; daily weight gain (DWG, g dia^−1^) = weight gain/experimental period; feed intake (FI, g) = feed consumption/final number of fish; daily feed intake (DFI, % dia^−1^) = daily feed consumption/weight gain × 100; feed conversion rate (FCR) = feed intake/weight gain; relative growth rate (RGR; % day^−1^) = (eg^−1^) × 100, where “e” is Euler’s number and g = (ln (final weight) − ln (initial weight))/(number of experimental days); condition factor (CF) = (body weight/total length^3^) × 100; batch uniformity in weight (BUW, %) = total number of animals with a total weight ± 10% within the average of each experimental unit/number of animals in the experimental unit × 100.

Blood samples were collected from the tail vein of three fish per tank, using 3 mL syringes rinsed with 5% EDTA anticoagulant, then the fish were subsequently euthanized for analysis of somatic indexes. Three more fish per experimental unit were used for fillet color analysis.

### 2.3. Somatic Indexes

Somatic indexes were estimated using the following formulas: viscero-somatic index (VSI, %) = (weight of the viscera/fish weight) × 100; hepatosomatic index (HSI, %) = (liver weight/fish weight) × 100; viscero-somatic fat index (VSFI, %) = (visceral fat weight/fish weight) × 100.

### 2.4. Hematological and Biochemical Parameters

The hemoglobin concentration (Hb) was determined using the cyanmethemoglobin method using a commercial kit (Labtest^®^, Vista Alegre, Lagoa Santa, MG, Brazil). The hematocrit (HT%) was determined using the microhematocrit method [24], the differential and total leukocyte and thrombocyte counts were obtained through an indirect method using blood smears stained with May-Grunwald-Giemsa [25]. Erythrocytes (RBC × 10^6^ cel µL^−^^1^) were counted in a hemocytometer (10 μL of blood, 2.0 mL of citrate formaldehyde). Corpuscular constants were determined using the Wintrobe methods [26] and using the following calculations: mean corpuscular hemoglobin concentration (MCHC, %) = [hemoglobin] × 100/hematocrit, mean corpuscular volume (MCV, fL) = hematocrit × 10/RBC and mean corpuscular hemoglobin (MCH, g dL^−^^1^) = [hemoglobin] × 10/RBC. The glucose level in the whole-blood was determined with a blood glucose meter (Accumed-Glicomed^®^, Rio de Janeiro, Brazil) immediately after blood collection [27].

Blood plasma was collected after centrifugation of the whole blood at 3000× *g* for 10 min at 4 °C. These samples were used to analyze the following biochemical variables according the protocol suggested by the manufacturer: total protein (TP), albumin (AB), globulin (GB), triglycerides (TG), cholesterol (CL), high-density lipoproteins (HDL), low-density lipoproteins (LDL), very low-density lipoproteins (VLDL) (In Vitro Diagnóstico, Itabira, Brazil) and total bilirubin (TB) (Labtest^®^, Vista Alegre, Lagoa Santa, MG, Brazil). Absorbance readings were taken in a spectrophotometer (HACH, model DR 6000, Loveland, CO, USA).

### 2.5. Fish Fillet Color

Three fish from each experimental unit were filleted (longitudinal muscular portion with a cut towards the backbone) and fillet color was measured immediately in the cranial, medial and caudal portion of the ordinary muscle with a colorimeter (Konica Minolta, CHROMA METER CR-200, Chiyoda, Japan) following the CIE system [28], thus recording L (brightness), a* (redness index) and b* (yellowing index). The average of the three portions was used to obtain the final result.

### 2.6. Statistical Analysis

All the data collected were tested for normality (Shapiro–Wilk) and homogeneity of variance (Levene) prior to further analysis. The diets were subjected to two groups of statistical analyses. In the first group, all the data from the diets, except for the control (CO), were subjected to regression analysis. The choice of regression models was based on the level of significance, the correlation coefficient (r^2^), and the best fit of the data for each variable based on Akaike’s criterion tests (AIC) and the F-test for equations from the same family of statistical distributions.

In the second group, all the diets were compared using one-way analysis of variance and, when significant, Dunn’s test. The diets containing the cassava by-products and the substitutions of fish meal for BSF meal were compared individually with the CO. Data that did not meet the assumptions of the analysis of variance were subjected to the Kruskal–Wallis test and the Dwass–Steel–Critchlow–Fligner (DSCF) median contrast test, and all the diets were compared pairwise. The significance level of *p* < 0.05 was used for all the analyses. Values are expressed as the mean ± standard deviation. The Statistica 13.0, CurveExpert Pro 2.7 and jamovi 2.3.18 programs were used to perform the analyses.

## 3. Results

The fish accepted all the experimental diets. The survival rate during the experimental period was 100%. No significant differences were observed for the variables FW, WG, DWG, RGR, CF and BUW (Table 4). Fish fed the control diet (CO) showed the best values for FCR and DFI. Fish fed 100BSFL presented higher value for FI compared to the fish fed 0BSFL and 25BSFL (Table 4). There was no difference for the whole-body proximate composition and biometric indices (Table 5 and Table 6).

There were no significant differences for the hematological parameters, with the exception of the Hb variable (Table 7). Fish fed 0BSFL showed lower Hb when compared to the control; however, there were no differences between them and the groups of fish fed diets containing BSFL. The plasma parameters GC, TP, AB, GB were similar for the experimental groups; nonetheless, TB was higher in fish fed 0BSFL when compared to the CO, 50BSFL, 75BSFL and 100BSFL, and was similar to the fish fed 25BSFL (Table 8). Fish fed 0BSFL had a higher plasma cholesterol content than those fed the CO diet (Table 9). There were no significant differences between the immunological parameters evaluated (Table 10).

The fish fillets presented similar values for the luminosity and redness among the experimental groups. The fillets of fish fed with the Amazonian diets (0BSFL, 25BSFL, 50BSFL, 75BSFL and 100BSFL) showed a higher level of yellowing than the fillets of fish that consumed the CO diet (Figure 2).

## 4. Discussion

Fish fed diets containing cassava by-products showed higher feed conversion rate values compared to fish fed with the control diet. Diets replacing fishmeal with BSFL meal (0BSFL to 100BSFL) presented a crude fiber content of 5.81 to 6.10% while the control diet contained 3.56%. Non-starch polysaccharides include cellulose, hemicellulose, pectic substances, gum and lignin [29] are often associated with a decrease in diet digestibility by fish [30,31]. This fact is due to the absence or scarcity of the enzymes β-glucanases and β-xylanases that digest fiber in freshwater teleosts [32]. The authors in [33] observed a decrease of 4.4% in the apparent digestibility coefficient (ADC) of total carbohydrates for Nile tilapia for every 1% increase in dietary fiber. In their natural habitats, the tambaqui feeds on fruits and seeds, with reports of levels of 9.7 to 20.5% of crude fiber in its stomach contents [34]. Although these levels are substantially higher than those found in the experimental diets of our study, it remains uncertain whether tambaqui possess the physiological capability to efficiently digest and utilize the high fiber content present in their natural food sources.

For tambaqui, the authors in [35] reported the ADC of dry matter, crude protein, and energy from cassava leaf meal of 49.43, 36.59% and 31.97%, respectively. These findings suggest that the cassava leaf meal contributed to reducing the feed conversion rate of fish fed diets formulated with cassava by-products (0 to 100BSFL). In addition to the high fiber content, cassava leaves contain anti-nutritional factors, such as tannins, phytate and cyanogenic acid [36], which can impair the digestibility and nutrient absorption, and harm the physiology of the fish [37]. The extrusion process is an efficient technique for reducing the tannin concentration of lentils by up to 98.83%, with enhanced results when the ingredient is hydrated (18%), when using a barrel temperature of 160 °C and a speed screw of 200 rpm. In the present study, the experimental diets were extruded after 28% hydration, at a temperature of 90 °C and 416 rpm, which may have contributed to the partial elimination of the anti-nutritional factors of the cassava by-products.

In tambaqui, the authors of [35] observed high ADC values for dietary cassava peel, i.e., an ADC of 88.52; 88.69; 81.73 and 89.09% for dry matter, protein, lipids and energy, respectively. In another study, the diet formulation used for tambaqui, which had the inclusion of up to 30% of cassava tuber residue meal replacing 100% of corn did not impact fish growth performance [38]. However, the fish fed with 30% cassava tuber residue meal presented a higher fat content in the muscle, when compared to the fish in the control group fed with a corn-based diet. This fact was not evidenced in the present study with the fish fed diets containing up to 19.98% cassava tuber residue meal plus 19.50% cassava peel meal.

Insects, a natural component of the tambaqui diet [3], possess chemical compounds that serve as attractants, enticing fish to consume them [39], which may be related to the higher feed intake of the 100BSFL diet. Moreover, the authors of [17] demonstrated that tambaqui efficiently digest defatted BSFL meal, with an ADC of 76, 84, 62, and 86% for dry matter, protein, lipids and energy, respectively. Although fishmeal and BSFL meal are well digested by the tambaqui, the amino acid composition of the diets must be considered, as unbalanced diets can adversely affect the fish growth, metabolism and immunity [40,41].

All the experimental diets contained the minimum lysine content (1.39%) for weight gain and feed conversion for tambaqui. However, the cassava by-products diets (0BSFL to 100BSFL) presented lysine levels that exceeded 1.58%, which, according to [42,43], is in line with the optimal thresholds for enhancing productivity and health in tambaqui. On the other hand, all the experimental diets presented a methionine + cystine content that was lower than the nutritional requirements recommended (0.86 to 0.93%), based on growth performance, body composition, erythogram, and the plasma and liver metabolites of tambaqui [44]. Therefore, the fish in the present study could have presented better production rates if supplementation of 0.14 to 0.20% of crystalline methionine had been used in all the experimental diets.

The dietary tryptophan content in cassava by-products was higher (0.58 to 0.61%) than the nutritional requirement value of 0.32% for juvenile tambaqui [45]; though these values were not sufficient to cause changes in the immunological parameters evaluated in the present study. Tryptophan is an essential amino acid that, in addition to participating in muscle building, plays a fundamental role in regulating neuroendocrine processes and the immune system of vertebrates [46]. However, excess tryptophan can induce oxidative stress in fish, with greater production of reactive oxygen species and low activities of antioxidant system enzymes (superoxide dismutase, catalase, glutathione peroxidase, glutathione-S-transferase and glutathione reductase), such as observed in the gills of grass carp (*Ctenopharyngodon idellus*) fed with a dietary deficiency or excess of tryptophan [47].

In relation to non-essential amino acids, in general, the diets (0 to 100BSFL) presented higher contents of glutamic acid, aspartic acid and glycine, when compared to the control diet (CO). These amino acids are recognized for their palatability [48], and are used as food stimulants in fish diets, especially in diets for carnivores that are prepared with a large proportion of vegetable ingredients [49]. An excess of non-essential amino acids (aspartic acid and glutamic acid) does not interfere with gluconeogenesis [50]. Glycine is the amino acid present in the greatest quantity in animals [48] and, therefore, is present in a high concentration in fish meal, which justifies its higher content in the 0BSFL and 25BSFL diets (100% and 75% of fish meal, respectively). Glycine plays an important role in the antioxidant system and ammonia detoxification in fish [46,51]. The glycine content of the experimental diets varied between 1.21 (100BSFL) and 3.56% (0BSFL), with the control diet (CO) having 2.15%. Common carp (*Cyprinus carpio*) fed diets supplemented with up to 3.01% glycine showed no alterations in growth performance values, but there were improvements in the antioxidant capacity of the fish [46].

Low-cost and non-lethal methods, such as hematological analysis, are an important tool for assessing the health status of fish in response to nutritional changes, water quality, and disease [52]. Hemoglobin is the protein responsible for transporting oxygen in the bloodstream, which during the physiological process is degraded into globin and the heme group. Both of which undergo further lysis and the globin portion is reused by the body while heme is converted to bilirubin [53]. Although the fish fed with the 0BSFL diet had a lower hemoglobin value (6.43 g dL^−^^1^) than the fish in the control group, this value does not interfere with the health of the fish, as this value is higher (5.61 g dL^−^^1^) than that found in tambaqui fed with a 15% inclusion of cassava tuber meal in the diet, and no negative impact on the health and growth performance of fish was reported [54].

The fish showed significant differences in the FCR, FI, DFI, Hb, CL and TB; however, no alterations were observed in the hematological, biochemical and immunological variables of tambaqui as a result of the use of the cassava by-products. The toxic compounds present in the cassava leaves, when properly neutralized, do not compromise the health of the fish. Therefore, the dehydration, grinding and cooking (extrusion) of feed during manufacturing were efficient in reducing the amount of toxic compounds (cyanogenic glycosides), and were sufficient to nullify their toxic effects in fish fed diets based on cassava by-products [11,12].

The chitin present in BSFL meal contributed to reducing cholesterol levels in the fish, which is in line with what was observed by [55]. Chitin, after being degraded by chitinase, is converted into chitosan, which can reduce cholesterol, as has already been observed in other experimental animal models [56]. In this study, the decrease in cholesterol in the tambaqui, due to the presence of chitin, can be explained both by causing hypocholesteronemia and by improving the immunity of the animals, especially in fish farming [57].

Cassava leaves contain carotenoids which contributed to the pigmentation of the fillets. The enhanced pigmentation was also observed in studies carried out with silver catfish (*Rhamdia quelen*) and carp (*Cyprinus carpio*) fed diets formulated with cassava leaves, and these showed a higher rate of yellowing of the fillets [58]. Dietary carotenoids act as a source of vitamins, provide liver protection and participate in processes in the response to stress and activity of the immune system [59]. Fish are not capable of synthesizing carotenoids, so they need to be supplied with carotenoids via the diet. In this sense, cassava leaf meal is a potential ingredient for supplying this nutrient in fish diets [60]. The alteration in the color of the tambaqui fillets could interfere with consumer acceptance of the fillets. Therefore, it is recommended to carry out future sensory analysis tests to assess consumer preferences.

## 5. Conclusions

Defatted BSF larvae meal and cassava by-products can be used to produce aquafeed for aquaculture in the Amazon. Further studies with a lower proportion of cassava leaves in the diet formulation are recommended so as to ensure enhanced diet digestibility and less impact on the color of the fillets.

## Figures and Tables

**Figure 1 animals-14-01165-f001:**
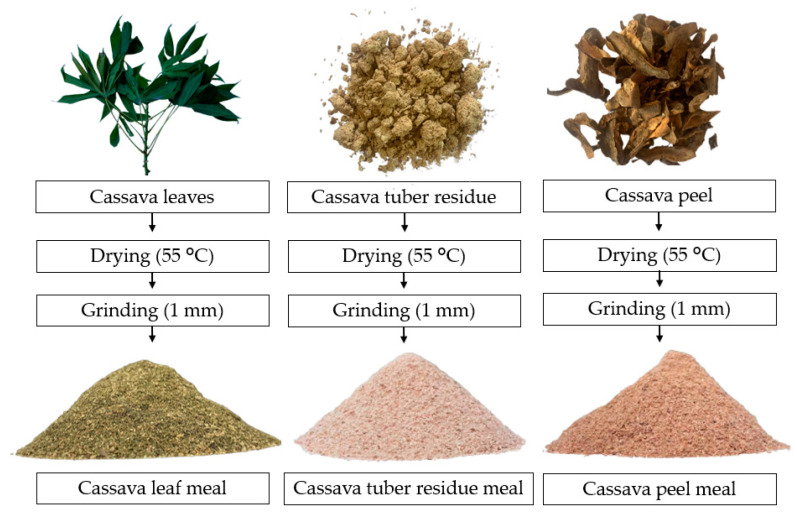
Flowchart of the processing of cassava by-products.

**Figure 2 animals-14-01165-f002:**
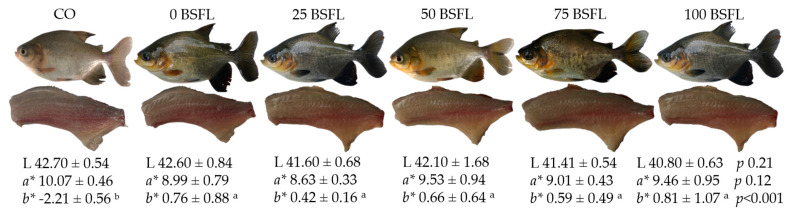
Fillet color of tambaqui (L) brightness, (*a**) redness index and (*b**) yellowing index, fed experimental diets for 60 days. One-way ANOVA *p*-values and different letters on the same line indicate significant difference (*p* < 0.05) between the diets via Dunn’s test.

**Table 1 animals-14-01165-t001:** Proximate composition and energy of the ingredients used in the preparation of the experimental diets.

Nutrients and Energy	Ingredients				
	CTM	CPM	CLM	BSFL	FM
Dry matter (%)	87.04 ± 0.46	90.24 ± 0.25	90.63 ± 0.23	89.78 ± 0.37	93.19 ± 0.32
Crude protein (%)	5.38 ± 0.52	6.12 ± 0.50	19.41 ± 0.51	59.65 ± 2.97	62.94 ± 0.53
Crude lipids (%)	4.19 ± 0.14	5.94 ± 0.38	8.05 ± 0.31	14.21 ± 0.75	8.41 ± 0.63
Crude fiber (%)	4.48 ± 0.34	8.01 ± 0.40	16.70 ± 0.42	0.40 ± 0.07	0.29 ± 0.02
Ashes (%)	3.46 ± 0.68	4.26 ± 0.32	3.60 ± 0.48	16.26 ± 0.37	21.61 ± 0.52
Gross energy (MJ kg^−1^)	18.29 ± 0.13	17.55 ± 0.05	17.08 ± 0.05	22.69 ± 0.25	18.01 ± 0.06

CTM: cassava tuber residue meal; CPM: cassava peel meal; CLM: cassava leaf meal; BSFL: defatted black soldier fly larvae meal; FM: whole-freshwater-fish meal.

**Table 2 animals-14-01165-t002:** Amino acid composition of the ingredients.

Amino Acid (g 100 g^−1^)	Ingredients		
	CLM	BSFL	FM
Essential amino acids			
Arginine	1.08	4.03	5.66
Histidine	0.42	2.41	1.35
Isoleucine	0.68	2.84	1.71
Leucine	1.30	4.34	3.71
Lysine	1.05	4.33	4.37
Methionine	0.24	1.11	1.40
Phenylalanine	0.96	2.52	1.83
Threonine	0.71	2.82	2.73
Tryptophan	0.89	1.02	1.05
Valine	0.95	4.10	2.20
Non-essential amino acids			
Alanine	1.06	3.66	5.13
Aspartic acid	3.09	5.68	4.98
Cystine	0.11	0.33	0.18
Glycine	0.92	2.63	8.68
Glutamic acid	3.33	7.10	8.69
Proline	0.85	3.67	4.70
Serine	1.23	2.90	3.02
Tyrosine	0.47	4.05	1.10
Taurine	0.07	0.10	0.43

CLM: cassava leaf meal; BSFL: defatted black soldier fly larvae meal and FM: fish meal.

**Table 3 animals-14-01165-t003:** Formulation and nutritional composition of the experimental diets.

Ingredients (g 100 g^−1^)	CO	Experimental Diets				
		0BSFL	25BSFL	50BSFL	75BSFL	100BSFL
Whole-freshwater-fish meal	-	38.50	28.88	19.25	9.62	0.00
Defatted BSFL meal	-	0.00	9.62	19.25	28.88	38.50
Cassava leaf meal	-	22.00	22.00	22.00	22.00	20.00
Cassava tuber residue meal	-	16.68	17.68	18.08	18.78	19.98
Cassava peel meal	-	16.00	17.00	18.00	18.00	19.50
Soybean meal	28.00	-	-	-	-	-
Meat and bone meal	18.50	-	-	-	-	-
Corn	33.48	-	-	-	-	-
Wheat middlings	13.50	-	-	-	-	-
Soybean oil	4.50	4.80	2.80	1.40	0.70	0.00
Vitamin/mineral supplement ^a^	1.00	1.00	1.00	1.00	1.00	1.00
Dicalcium phosphate	0.50	0.50	0.50	0.50	0.50	0.50
Salt	0.50	0.50	0.50	0.50	0.50	0.50
BHT	0.02	0.02	0.02	0.02	0.02	0.02
Nutrients and energy						
Crude protein (g 100 g^−1^)	28.175	28.012	28.098	28.152	28.162	28.013
Corrected protein (g 100 g^−1^) ^b^	28.175	28.012	27.501	26.966	26.376	25.635
Crude lipids (g 100 g^−1^)	8.268	11.432	10.106	9.346	9.237	9.079
Ash (g 100 g^−1^)	10.474	11.352	10.912	10.450	9.961	9.483
Crude fiber (g 100 g^−1^)	3.564	5.816	5.950	6.058	6.100	5.959
Starch (g 100 g^−1^)	25.542	26.366	27.838	28.824	29.395	31.167
Gross energy (MJ kg^−1^)	18.443	18.436	18.455	18.614	18.910	19.237
Essential amino acids (g 100 g^−1^) ^c^						
Arginine	1.85	2.47	2.32	2.16	2.01	1.83
Histidine	0.61	0.64	0.74	0.85	0.95	1.05
Isoleucine	0.99	0.84	0.95	1.06	1.17	1.27
Leucine	1.91	1.77	1.83	1.89	1.96	1.99
Lysine	1.39	1.95	1.95	1.95	1.94	1.92
Methionine	0.37	0.60	0.58	0.55	0.52	0.49
Phenylalanine	1.10	0.96	1.02	1.09	1.16	1.21
Threonine	0.95	1.24	1.25	1.26	1.27	1.26
Tryptophan	0.27	0.61	0.61	0.61	0.60	0.58
Valine	1.20	1.10	1.28	1.47	1.65	1.82
Non-essential amino acids (g 100 g^−1^) ^c^						
Alanine	1.58	2.21	2.07	1.93	1.78	1.62
Aspartic acid	1.43	2.60	2.66	2.73	2.80	2.80
Cystine	0.35	0.11	0.12	0.14	0.15	0.17
Glycine	2.15	3.56	2.98	2.40	1.81	1.21
Glutamic acid	2.19	4.08	3.93	3.77	3.62	3.40
Proline	1.82	2.00	1.90	1.80	1.70	1.58
Serine	1.25	1.45	1.44	1.42	1.41	1.38
Tyrosine	0.79	0.55	0.83	1.12	1.40	1.68
Taurine	0.00	0.18	0.15	0.12	0.09	0.05

Abbreviations: BSFL: defatted black soldier fly larvae; BHT: Butylhydroxytoluene. ^a^ Vitamin and mineral mix (Nutron^®^) per kg of product: folic acid (250 mg), pantothenic acid (5000 mg), antioxidant (600 mg), biotin (125 mg), cobalt (25 mg), copper (2000 mg), iron (13,820 mg), iodine (100 mg), manganese (3750 mg), niacin (5000 mg), selenium (75 mg), vitamin A (1000.000 IU), vitamin B1 (1250 mg), vitamin B12 (3750 mg), vitamin B2 (2500 mg), vitamin B6 (2485 mg), vitamin C (28,000 mg), vitamin D3 (500,000 IU), vitamin E (28,000 IU), vitamin K3 (500 mg), and zinc (17,500 mg). ^b^ Corrected protein was calculated using the correction factor 5.6 [19]. ^c^ Calculated based on chemical analysis of the amino acid composition of cassava leaf meal, defatted black soldier fly larvae meal, whole-freshwater-fish meal and, for the other ingredients, data available in [21] were used.

**Table 4 animals-14-01165-t004:** Growth performance of juvenile tambaqui fed the experimental diets or the control diet for 60 days.

Diets	FW (g)	WG (g)	DWG (g dia^−1^)	FCR	FI (g)	DFI (% dia^−1^)	RGR (% dia^−1^)	CF	BUW (%)
CO	71.44 ± 7.53	46.83 ± 7.53	0.78 ± 0.12	1.33 ± 0.06 ^b^	62.06 ± 7.73 ^ab^	2.21 ± 0.11 ^b^	1.82 ± 0.19	1.80 ± 0.06	48.75 ± 11.09
0BSFL	61.04 ± 4.22	36.43 ± 4.21	0.61 ± 0.07	1.70 ± 0.19 ^a^	61.31 ± 1.95 ^b^	2.80 ± 0.22 ^a^	1.55 ± 0.12	1.83 ± 0.03	43.75 ± 13.15
25BSFL	60.45 ± 6.58	35.84 ± 6.58	0.60 ± 0.11	1.69 ± 0.06 ^a^	60.35 ± 8.73 ^b^	2.81 ± 0.11 ^a^	1.53 ± 0.18	1.81 ± 0.05	45.00 ± 5.00
50BSFL	60.42 ± 7.91	35.81 ± 7.91	0.60 ± 0.13	1.84 ± 0.15 ^a^	64.98 ± 9.54 ^ab^	2.98 ± 0.28 ^a^	1.52 ± 0.23	1.77 ± 0.06	41.67 ± 10.41
75BSFL	61.03 ± 4.47	36.42 ± 4.47	0.61 ± 0.07	1.81 ± 0.16 ^a^	65.23 ± 2.89 ^ab^	2.99 ± 0.21 ^a^	1.55 ± 0.13	1.81 ± 0.08	33.75 ± 7.50
100BSFL	68.90 ± 6.30	44.29 ± 6.30	0.74 ± 0.10	1.77 ± 0.12 ^a^	77.93 ± 5.48 ^a^	2.93 ± 0.23 ^a^	1.76 ± 0.15	1.86 ± 0.03	41.67 ± 20.21
*p*	0.09	0.09	0.09	0.02	0.03 *	0.04	0.10	0.38	0.63
Regression									
*p*	ns	ns	ns	ns	0.04	ns	ns	ns	ns
r^2^					0.49				
Model					Quadratic				

CO: control diet; FW: final weight; WG: weight gain; DWG: daily weight gain; FCR: feed conversion rate; FI: feed intake; DFI: daily feed intake; RGR: relative growth rate; CF: condition factor; BUW: batch uniformity in weight. * Kruskal–Wallis test and Dwass–Steel–Critchlow–Fligner median contrast test. One-way ANOVA *p*-values and different superscript letters in the same column indicate a significant difference (*p* < 0.05) between the diets via Dunn’s test. ns = not significant (*p* > 0.05).

**Table 5 animals-14-01165-t005:** Proximate composition of the whole body of juvenile tambaqui fed the experimental diets or the control diet for 60 days.

Diet	DM (%)	CP (%)	CL (%)	AS (%)	GE (MJ kg^−1^)
CO	30.18	14.90	6.46	3.38	1454.21
0BSFL	30.08	15.14	6.02	3.76	1409.79
25BSFL	29.91	15.50	6.21	3.43	1387.59
50BSFL	28.99	14.03	5.51	3.14	1293.18
75BSFL	29.79	14.42	6.40	3.45	1442.15
100BSFL	28.83	14.93	5.78	3.34	1376.57
Pooled SE	0.89	0.94	0.39	0.24	0.24
*p*	0.07	0.19	0.37	0.32	0.07 *
Regression					
*p*	ns	ns	ns	ns	ns

CO: control diet; DM: dry matter; CP: crude protein; CL: crude lipids; AS: ash; GE: gross energy; SE: Standard Error. * Kruskal–Wallis test and Dwass–Steel–Critchlow–Fligner median contrast test. One-way ANOVA *p*-values (*p* < 0.05) indicate a significant difference between the diets. ns = not significant (*p* > 0.05).

**Table 6 animals-14-01165-t006:** Somatic indexes of juvenile tambaqui fed the experimental diets or the control diet for 60 days.

Diet	VSI (%)	HSI (%)	VSFI (%)	SSI (%)
CO	7.91	2.13	1.97	0.10
0BSFL	7.48	1.77	1.99±	0.07
25BSFL	7.64	2.08	1.50	0.06
50BSFL	7.49	1.87	1.37	0.07
75BSFL	8.21	2.25	1.81	0.07
100BSFL	8.26	2.28	1.91	0.10
Pooled SE	0.54	0.29	0.34	0.03
*p*	0.21	0.13	0.11	0.27
Regression				
*p*	ns	0.007	ns	ns
r^2^		0.39		
Model		Linear		

CO: control diet; VSI: viscero-somatic index; HSI: hepatosomatic index; VSFI: viscero-somatic fat index; SSI: splenic-somatic index; SE: Standard Error. One-way ANOVA *p*-values (*p* < 0.05) indicate a significant difference between the diets. ns = not significant (*p* > 0.05).

**Table 7 animals-14-01165-t007:** Hematological parameters of juvenile tambaqui fed the experimental diets or the control diet for 60 days.

Diets	HT (%)	Hb (g dL^−1^)	ET (10^6^ µL^−1^)	MCV (fL)	MCH (pg)	MCHC (g dL^−1^)
CO	33.84	8.13 ^a^	1.50	227.53	54.70	24.06
0BSFL	28.58	6.43 ^b^	1.25	231.04	51.70	22.66
25BSFL	28.52	6.63 ^ab^	1.40	212.00	48.90	23.30
50BSFL	32.05	6.67 ^ab^	1.44	218.00	47.10	20.84
75BSFL	32.44	7.22 ^ab^	1.33	249.00	55.70	22.30
100BSFL	33.52	7.45 ^ab^	1.43	248.00	54.50	22.27
Pooled SE	2.48	0.72	0.12	27.44	6.95	1.76
*p*	0.07	0.04 *	0.12	0.22	0.47	0.31
Regression						
*p*	ns	ns	ns	ns	ns	ns

CO: control diet; HT: hematocrit; Hb: hemoglobin; ET: erythrocytes; MCV: mean corpuscular volume; MCH: mean corpuscular hemoglobin; MCHC: mean corpuscular hemoglobin concentration; SE: Standard Error. * Kruskal–Wallis test and Dwass–Steel–Critchlow–Fligner median contrast test. One-way ANOVA *p*-values and different superscript letters in the same column indicate a significant difference (*p* < 0.05%) between the diets via Dunn’s test. ns = not significant (*p* > 0.05%).

**Table 8 animals-14-01165-t008:** Biochemical parameters of juvenile tambaqui fed the experimental diets or the control diet for 60 days.

Diet	GC (mg dL^−1^)	TP (g dL^−1^)	AB (g dL^−1^)	GB (g dL^−1^)	TB (mg dL^−1^)
CO	88.80	3.93	0.86	3.11	23.10 ^b^
0BSFL	88.30	4.18	1.15	3.03	32.80 ^a^
25BSFL	89.90	4.38	0.87	3.52	26.83 ^ab^
50BSFL	89.00	3.75	1.28	2.48	21.75 ^b^
75BSFL	97.30	3.99	0.91	3.07	23.27 ^b^
100BSFL	86.00	3.40	0.92	2.63	24.60 ^b^
Pooled SE	16.99	0.63	0.25	0.66	3.84
*p*	0.94	0.49	0.24	0.46	0.01
Regression					
*p*	ns	ns	ns	ns	ns

CO: control diet; GC: glucose; TP: total protein; AB: albumin; GB: globulin; TB: total bilirubin; SE: Standard Error. Data were analyzed using one-way ANOVA. Different letters in the same column indicate significant difference (*p* < 0.05) between the diets via Dunn’s test. ns = not significant (*p* > 0.05).

**Table 9 animals-14-01165-t009:** Blood lipid profile of juvenile tambaqui fed the experimental diets or the control diet for 60 days.

Diet	TG (mg dL^−1^)	CL (mg dL^−1^)	HDL (mg dL^−1^)	LDL (mg dL^−1^)	VLDL (mg dL^−1^)
CO	167.08	85.34 ^b^	25.03	27.10	33.40
0BSFL	152.78	107.00 ^a^	36.13	40.60	30.60
25BSFL	153.00	95.30 ^ab^	28.04	28.90	33.10
50BSFL	174.07	97.10 ^ab^	26.57	35.00	35.10
75BSFL	198.29	95.00 ^ab^	28.10	32.80	39.70
100BSFL	177.86	93.40 ^ab^	28.03	32.00	35.40
Pooled SE	33.65	7.72	4.19	8.31	6.73
*p*	0.46	0.007 *	0.20 *	0.26 *	0.46 *
Regression					
*p*	ns	ns	ns	ns	ns

CO: control diet; TG: triglycerides; CL: cholesterol; HDL: high-density lipoproteins; LDL: low-density lipoproteins; VLDL: very low-density lipoproteins; SE: Standard Error. * Kruskal–Wallis test and Dwass–Steel–Critchlow–Fligner median contrast test. Data were analyzed using one-way ANOVA. Different letters in the same column indicate significant difference (*p* < 0.05) between the diets via Dunn’s test. ns = not significant (*p* > 0.05).

**Table 10 animals-14-01165-t010:** Immunological parameters (leukocytes and thrombocytes) of juvenile tambaqui fed the experimental diets for 60 days.

Diets			(10^3^ µL^−1^)		
	TL	LP	MO	NE	TB
CO	113.84	92.59	15.56	5.59	21.01
0BSFL	104.12	88.01	12.46	3.45	13.53
25BSFL	108.32	88.94	16.28	3.05	15.46
50BSFL	117.18	98.20	14.44	4.22	18.68
75BSFL	120.77	101.30	15.55	3.92	20.40
100BSFL	116.78	92.70	17.32	6.56	22.33
Pooled SE	11.88	11.84	4.71	2.16	5.80
*p*	0.37 *	0.62	0.80	0.32	0.33
Regression					
*p*	ns	ns	ns	ns	ns

CO: control diet; TL: total leukocytes; LP: lymphocytes; MO: monocytes; NE: neutrophils; TB: thrombocytes; SE: Standard Error. * Kruskal–Wallis test and Dwass–Steel–Critchlow–Fligner median contrast test. Data were analyzed using one-way ANOVA.

## Data Availability

Data from this study are available from the corresponding authors upon reasonable request.

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
