# Peer review of "A Sustainable Diet for Tambaqui Farming in the Amazon: Growth Performance, Hematological Parameters, Whole-Body Composition and Fillet Color"

_animals, 2024, doi:10.3390/ani14081165_

Round 1

Reviewer 1 Report

Comments and Suggestions for Authors

Reviewer 2 Report

Comments and Suggestions for Authors

This MS reports on a study examining the use of cassava byproducts and black soldier fly larvae meal (BSFL) in diets for tambaqui. The study contains information that could be useful for those interested in the culture of these animals. The experimental design and statistical analysis are suitable for this type of study. The conclusions are supported by the results. The writing is clear and understandable.

The authors should consider the following minor points:

- Figure 2, radar chart. I always have trouble reading radar charts for amino acids, as the values are not intuitively obvious to me. My preference is to present these data in tabular format. This is just my opinion.

- Line 170. Where were the fish obtained from, what age were they and what was their dietary history?

- Line 194, PER. If the diets were iso-proteinaceous (Line 149), then there is no need to have both FCR and PER, as they are just a scalar (and inverted) different.

- Given that there was no significant difference in weight gain (Table 3) among dietary treatments, but a significant linear influence in feed intake, it is surprising that there is no significant difference in FCR, excluding the control diet. I wonder if feed efficiency (WG/FI) would be more statistically sensitive in this way. 

- For the tables (except Table 3), it is more statistically defensible to use pooled standard errors for parameter, rather than individual means, as long as the equal variance test (Levene) was passed. In other words, if the variances were found to be equal, then there's no statistical reason to split them into individual means.

- It is unfortunate that the experimental diets were deficient in Met+Cys (sulfer-containing AA), and this could be the result of the poor performance, and the lack of dietary effect. Still, the authors draw attention to this in Lines 414-419 and the reader can decide. 

- Table 4. The proximate composition of the fish should be presented on wet weight basis, not dry weight. This will not change any conclusions, but it will be a better description of the impact of the treatments on the fish composition.

- Line 417. "could have possibly" may be shortened to "could have".

- Line 441-442. "... diets supplemented with up to 3.01% glycine showed..."

Reviewer 3 Report

Comments and Suggestions for Authors

The societal need for the study is well described and the study is thus of interest for readers.

Production and chemical composition of the test ingredients is adequately described. Presenting the amino acid profile of the ingredients with a diagram is interesting and even ok, although many readers might want to see the values in table. The diet formulations seem to be ok, although levels of the cassava products do not differ very much between the experimental diets.

In Table 2 I would suggest presenting the crude nutrient contents with three numbers: two digits for example for protein and lipids is pseudo-accuracy, beyond analytical accuracy.

Regarding the conditions in the feeding trial, although water quality seems to be ok, there is a need to describe how much new water was put in the systems on a daily basis. Few words about the phytoremediation (line 172) should be included. Fish were fed until satiation for 60 days, during which time fish grew from 25 grams to 60-70 grams, thus appr. four folding the initial weight which is adequate for a growth assay. No fish died, which further indicates that conditions were good enough for the fish.

Proximate composition should be presented on as-is basis, since it is the standard way of expressing proximate composition and because it allows readers to calculate values such as protein and lipid retention.

Instead of using RGR, please replace it with SGR (specific growth rate, the standard parameter in aquaculture studies) since I understood it is calculated similarly.

I didn’t find any major weaknesses in the study. Authors already note, that lower contents of cassava should be studied in the future, and such treatments could have been incorporated in this study already.
